# Redox Metabolism Measurement in Mammalian Cells and Tissues by LC-MS

**DOI:** 10.3390/metabo11050313

**Published:** 2021-05-13

**Authors:** Boryana Petrova, Anna Warren, Nuria Yulia Vital, Andrew J. Culhane, Adam G. Maynard, Alan Wong, Naama Kanarek

**Affiliations:** 1Department of Pathology, Boston Children’s Hospital, 300 Longwood Avenue, Boston, MA 02115, USA; boryana.petrova@childrens.harvard.edu (B.P.); awarren3@conncoll.edu (A.W.); nuria.vital@mail.huji.ac.il (N.Y.V.); andrew.culhane@childrens.harvard.edu (A.J.C.); adam.maynard@childrens.harvard.edu (A.G.M.); alan.wong@childrens.harvard.edu (A.W.); 2Harvard Medical School, Boston, MA 02115, USA; 3Program in Biological and Biomedical Sciences, Harvard Medical School, Boston, MA 02115, USA; 4Broad Institute of Harvard and Massachusetts Institute of Technology, 415 Main Street, Cambridge, MA 02142, USA

**Keywords:** redox metabolites, mass-spectrometry method, HILIC chromatography, NADH, NADPH, glutathione, redox metabolite detection in mammalian cells

## Abstract

Cellular redox state is highly dynamic and delicately balanced between constant production of reactive oxygen species (ROS), and neutralization by endogenous antioxidants, such as glutathione. Physiologic ROS levels can function as signal transduction messengers, while high levels of ROS can react with and damage various molecules eliciting cellular toxicity. The redox state is reflective of the cell’s metabolic status and can inform on regulated cell-state transitions or various pathologies including aging and cancer. Therefore, methods that enable reliable, quantitative readout of the cellular redox state are imperative for scientists from multiple fields. Liquid-chromatography mass-spectrometry (LC-MS) based methods to detect small molecules that reflect the redox balance in the cell such as glutathione, NADH, and NADPH, have been developed and applied successfully, but because redox metabolites are very labile, these methods are not easily standardized or consolidated. Here, we report a robust LC-MS method for the simultaneous detection of several redox-reactive metabolites that is compatible with parallel global metabolic profiling in mammalian cells. We performed a comprehensive comparison between three commercial hydrophilic interaction chromatography (HILIC) columns, and we describe the application of our method in mammalian cells and tissues. The presented method is easily applicable and will enable the study of ROS function and oxidative stress in mammalian cells by researchers from various fields.

## 1. Introduction

Cellular redox could be defined as the highly dynamic and tightly regulated balance between molecules that function as oxidants and antioxidants. Cells constantly produce reactive oxygen species (ROS) through partial reduction of O_2_ by the electron transport chain (ETC). ROS are inactivated by enzymes such as catalase, and endogenous metabolites that function as antioxidants, such as glutathione. ROS have a role in signal transduction [1] and are therefore important for normal physiology [2]. However, if ROS levels exceed their tightly-controlled normal levels, they induce oxidative stress, and can cause damage to proteins, lipids, and DNA [3]. Thus, redox state must be finely tuned to match the specific metabolic needs of the cell or tissue.

The various roles played by ROS in physiological and pathological conditions draw the attention of researchers from diverse biological fields, including cancer [4,5], immunology [6,7], neurobiology [8], aging [9], and more. To study ROS, either in the context of ROS as signalling molecules, or as a cause for oxidative stress, it is imperative to be able to experimentally measure cellular ROS levels, and to portray the cellular redox state. Measuring ROS is a long-standing challenge [10,11,12], and is still an active area of investigation. Measurement of ROS should be carefully done as each marker can be very cell-specific and biological variability could be significant among different conditions and biological systems [11]. Approaches can be broadly divided into three categories: measuring ROS directly, measuring the oxidative damage that results from high ROS, or assessing the metabolic redox state by determining reduced and oxidized redox-reactive small molecule pools. Several probe-assisted methods have been developed recently that measure a distinct ROS species by either a genetically encoded sensor or by positron emission tomography (PET) probe tracing [13,14,15,16,17,18,19,20].

However, for many researchers, the global cellular redox state is the more relevant readout of oxidative balance because it is the sum of oxidizing and reducing potential that dictates cell fitness and functionality [5]. Cellular redox state can be assessed by profiling the ratio of reduced and oxidized redox-reactive small molecules such as reduced- to oxidized-glutathione (GSH/GSSG), nicotinamide adenine dinucleotide (NAD) to reduced NAD (NAD^+^/NADH), nicotinamide adenine dinucleotide phosphate (NADP) to reduced NADP (NADP^+^/NADPH), the abundance of the oxidized form of the amino acid cysteine—cystine, and homocysteine and its precursor—cystathionine. These small molecules are directly accessible by liquid-chromatography coupled to mass-spectrometry (LC-MS). Indeed, LC-MS-based detection of these metabolites has been successfully applied in mammalian cells and organisms and revealed very impactful new biology in various fields [21,22,23,24]. However, by nature, redox-reactive metabolites are very labile; therefore, their extraction and detection are challenging, and results can be misleading. Furthermore, improvement of direct LC-MS quantitation has lagged. As more and more labs incorporate metabolite profiling into their experimental approaches, we recognized a need for a robust protocol for consistent detection of redox-reactive metabolites that can be done in parallel to global metabolite detection methods. This can allow for comprehensive profiling of redox and non-redox metabolites in the same experiment and provide information about the general metabolic state of the cell side by side with its redox state. 

Here we describe a LC-MS method for detection of redox-reactive metabolites (hereafter; redox metabolites) in parallel to global metabolic profiling in mammalian cells. We have performed a comprehensive comparison between three commercially available chromatography columns; Accucore Amide HILIC (Thermo-Fisher Scientific, Waltham, MA, USA), LUNA-NH2 (Phenomenex, Torrance, CA, USA) or SeQuant ZIC-pHILIC (Millipore-Sigma, St. Louis, MI, USA), and optimized extraction method for mammalian cells, tissues, and physiological fluids. Our protocol and data provide guidance for optimization of the LC-MS method to allow for tailored metabolite profiling of redox metabolites in parallel to other metabolites of interest.

## 2. Results

### 2.1. A Method to Quantify Redox Metabolites by LC-MS

To address the need for quantification of redox metabolites in parallel to detection of polar metabolites from mammalian cells we set to optimize an LC-MS method based on hydrophilic interaction chromatography (HILIC). In addition to the well-recognized SeQuant ZIC-pHILIC columns (Millipore-Sigma, St. Louis, MI, USA) we decided to explore two more consistently available columns: Accucore-Amide HILIC (Thermo-Fisher Scientific, Waltham, MA, USA) and LUNA-NH2 (Phenomenex, Torrance, CA, USA). We first optimized chromatographic solvents, gradient, and temperature based on literature [25,26] and aiming for a broad metabolite coverage (generally in the basic pH range). When comparing the chromatography and detection of several common intracellular metabolites such as amino acids, nucleotides, lactate, and glucose, we observed variation in ionization efficiency of these metabolites, as well as varying peak widths and retention times (Figure 1a and Table 1). The observed subtle differences in retention times of the measured metabolites are expected, due to the different gradient conditions and column chemistry in the three tested columns. However, because most metabolites were well detected via all three columns, we decided to use all three for optimization of the redox metabolites detection method.

Of note, we have tried to detect redox metabolites using the C18 column (Ascentis Express C18 HPLC column (2.7 μm × 15 cm × 2.1 mm; Sigma Aldrich, St. Louis, MI, USA)) but were not successful. Additionally, the use of HILIC columns allows detection of redox metabolites in parallel to global polar metabolite detection that is a major advantage of this chromatography over others. 

We focused our efforts on GSH, GSSG, NADH, NAD^+^, NADPH, and NADP+, because these metabolites are critical for redox balance and are most commonly assessed as a readout for the cellular redox state. As NAD^+^ and NADP^+^ are in the same molecular weight range as well as significantly more stable than their reduced counterparts, we excluded them from our initial optimization of mass spectrometry parameters (Figure 1b,c). Conversely, we assessed optimization parameters and the detection efficiency of both native GSH and GSH derivatized with Ellman’s reagent (5,5′-dithiobis-2-nitrobenzoic acid) [27] (abbreviated GSH-Ell; Appendix A). We were capable to reliably and reproducibly detect all tested redox metabolite standards on all three HILIC columns, although sensitivity varied among the columns (Figure 1b). We operate a Thermo QE mass spectrometer under heated electrospray ionization (HESI) conditions. To optimize detection by MS of the redox metabolite standards we tested various HESI source parameters (Figure 1c and Appendix A) and compared the limit of detection and linearity (Figure 1d and Table 2). We observed linear detection for almost all redox metabolites in all three columns with the exception of GSH-Ell on the LUNA-NH2 column. Notably, compared to GSH and other small molecular weight standards (Appendix A), GSH-Ell has HESI parameters closer to NAD(P)H and GSSG while at the same time a limit of detection similar to GSH on ZIC-HILIC and Accucore columns, offering the possibility to combine optimum parameters for the detection of all relevant redox reactive pairs. We conclude that optimal HESI conditions for small polar metabolites and tested redox metabolites are T-capillary ≥ 350 °C ≤ 400 °C and S-lens ≥ 60. In summary, following method optimization, all three columns tested here are suitable for detection of NADH, NADPH, GSH, and GSSG and our protocol can be applied directly by laboratories that work with any of these chromatography columns. 

### 2.2. Optimization of Mammalian Cell Sample Preparation for Redox Metabolite Detection

Following the reliable and sensitive detection of redox metabolites by LC-MS, we set out to optimize extraction solvent and storage conditions for detection of the endogenous metabolites in mammalian cells. We tested three extraction buffers for optimal metabolite extraction from the mammalian cell line K562. The buffers A, B, and C (Table 3) were used in an identical extraction protocol and analyzed using the ZIC-pHILIC column for LC-MS.

We used three increasing amounts of cells to demonstrate the reproducibility and linearity of the results (Table 4). 

Buffer A appeared sub-optimal for detection of NADPH, while buffer B was suboptimal for GSSG, NADPH, and NADP+ detection (Figure 2a). Buffer C was suitable for detection of GSH, GSSG, NAD^+^, NADH, and NADP^+^, but not NADPH. Of note are the lower levels of GSSG in Buffer C and C+Ell; oxidation of GSH to GSSG and GSSG extraction efficiency are confounding factors when it comes to the total measured GSSG levels, however, the presence of ascorbate in the buffer (as in buffers B and C) as well as Ellman’s reagent (as in buffer C+Ell) likely prevents oxidation of GSH and thus lead to lower levels of detected GSSG. Further, the addition of the Ellman’s reagent to buffer C resulted in reproducible detection of GSH-Ell as well as of NADPH, likely as a result of the stabilization of GSH and its indirect effect on stabilizing NADPH levels [30] (Figure 2a). All buffers were suitable for detection of other metabolites such as amino acids (Appendix A). We further tested the stability of endogenous redox metabolites in mammalian cell extracts using extraction buffers B and C at various storage conditions (Figure 2b). We observed no significant decline in signal with storage at −80 °C and the additional storage at 4 °C (4 or 24 h) for amino acids (Appendix A) and for the redox metabolites we measured, with the exception of an increase in GSSG in buffer C, indicative of measurable oxidation of GSH. In addition, small and not significant changes were observed in GSH, NADH, and NADPH in buffer B, of NADH in buffer C+Ell, and NADPH in buffer C after 24 h at 4 °C (Figure 2c). These inconsistent and insignificant changes can be the result of chemical interconversion upon storage, and noise due to low detection level. In summary, buffer C+Ell showed optimal in terms of extraction efficiency, storage stability and linearity of detection for endogenous redox metabolites in K562 cells. Adaptations of the buffer composition showed fruitful in enhancing detection of specific metabolites, and therefore should be applied according to specific interest in one metabolite or other. We conclude that buffer C+Ell should be adopted as the default buffer for researchers who are interested in measuring redox metabolites in mammalian cells in parallel to detection of other polar metabolites.

### 2.3. Application of the Redox Metabolite Detection Method for Mammalian Tissues and Biofluids

Next, we applied our method for detection of redox metabolites in parallel to global metabolic profiling of mammalian tissues. For solid organs, we focused on mouse liver and kidney, because we wanted to test our method in organs with high metabolic complexity. We compared the detection of 215 polar metabolites by the three HILIC columns: Accucore Amide HILIC, LUNA-NH2, and ZIC-pHILIC in mouse liver (Figure 3a and Appendix A), and kidney (Figure 3b and Appendix A). We observed significant differences between these columns, with specific metabolites detected better in each column, where ZIC-pHILIC had the broadest coverage (Figure 3c and Appendix A, Table 5). The reported results here can guide researchers in selecting preferred column based on their specific metabolites of interest.

We also compared the detection of polar metabolites following extraction of mouse liver using three extraction buffers: B, C, and C + Ellman’s (Appendix A). We observed no significant difference between these extraction buffers overall (Appendix A), with the exception of few specific metabolites that were detected better in each buffer (Figure 3d,e). Redox metabolites; GSH, GSSG, NAD^+,^ NADH, and NADP^+^ were well detected in mouse liver and kidney, while NADPH was well detected in liver only (Figure 3f). However, we were able to detect other metabolites that reflect the oxidative state of the cells in mouse kidney. These include cystine, that is the oxidized form of the amino cysteine, and cystathionine, a homocysteine precursor (Figure 3f). Next, we tested the detection of redox metabolites in two mouse body fluids: the cerebrospinal fluid (CSF) and plasma. We were able to detect oxidized and reduced glutathione in mouse CSF extracted in buffer C and in buffer C+Ell, as well as GSH-Ellman’s. The detection of GSH in the CSF was more consistent and reliable when the sample was extracted in buffer C plus Ellman’s reagent, likely due to high oxidation rate of metabolites in the CSF. We detected no GSSG in the plasma, but GSH-Ellman’s was detected in buffer C+Ell, as well as cystine and cystathionine in mouse plasma extracted in either extraction buffer (Figure 3f). Other metabolites, such as amino acids, were well-detected in all three buffers in all analysed tissues (Appendix A). We found that liver GSH is optimally derivatized with as little as 20 μmol Ellman’s reagent (Appendix A). Sample drying and reconstitution can help concentrate low abundance metabolites when small amounts are analyzed. However, given the high abundance of metabolites in liver samples, we asked if we could omit these steps prior to LC-MS analysis. This could have the added advantage of speeding up the preparation time and minimizing exposure to oxygen. In mouse liver samples, detection of NADH and NADPH was indeed improved when we did not apply drying of the sample (Appendix A). We conclude from these results that our LC-MS method is adequate for the detection of redox metabolites in parallel to global metabolic profiling of mammalian tissues. Broadest metabolite coverage and least redox metabolite oxidation is achieved when combining ZIC-pHILIC chromatography with buffer C+Ell extraction, while not compromising the detection of the vast majority of other polar metabolites. In addition, when feasible, skipping the drying step can further prevent oxidation of redox metabolites and improve detection of metabolites of interest. 

### 2.4. Application of the Redox Metabolite Detection Method for Profiling Redox State Following Pharmacologic Perturbations of Redox Balance in Cells

We wanted to test our redox metabolite detection method in mammalian cells following perturbation of the redox state. To that end, we treated cells with either the reactive oxygen species H_2_O_2_, the oxidizing agent, diamide (DA) [31], or with oligomycin (OM)—an inhibitor of ATP synthase [32] that inhibits the electron transport chain (ETC), and causes accumulation of NADH. We also monitored cellular redox balance following treatment with the drug methotrexate, which was shown to cause oxidative damage in mammalian tissues [33,34,35] and that is commonly used as a chemotherapeutic agent for the treatment of blood cancers [36]. Importantly, our method allowed us to profile, in parallel, the cell’s global metabolome in response to the treatments (Figure 4a and Appendix A) and to verify that the applied treatments resulted in the predicted changes in the metabolic profile of the cells (Appendix A). As expected, H_2_O_2_, DA, and OM treatments resulted in a decreased GSH/ GSSG ratio due to the oxidative conditions they induce in the cells, treatment with the oxidizing agents H_2_O_2_ and DA resulted in decreased NADH, and ETC inhibition by OM resulted in accumulation of NADH. NADPH levels increased following all treatments, indicating the anti-oxidative response of the cells to all treatments [37,38] (Figure 4b). These results emphasize our ability to detect the high adaptability of the cancer cells we tested and their dynamic response to oxidative stress. In conclusion, our optimized method based on ZIC-pHILIC-chromatography and buffer C+Ell extraction provides the opportunity to comprehensively study the metabolic consequences of acute perturbations in mammalian cells by combining redox and metabolic state measurements in a single LC-MS measurement.

## 3. Discussion

Here we report the optimization of an LC-MS detection method of redox metabolites suitable for parallel assessment of global metabolism from mammalian cells, tissues, and body fluids. This method will be of interest for researchers from a wide variety of biological fields. We describe optimized detection of chemical standards of the redox metabolites, as well as optimized detection of the endogenous metabolites in mammalian cells and in mouse tissues—liver, kidney, CSF, and plasma. Further, we measured the levels of redox metabolites in the cancer cell line K562 following pharmacological perturbation of the cellular redox state and demonstrate our ability to assess relevant metabolites in cells exposed to mild oxidative stress.

Our data suggest that optimal detection of several redox metabolites can be achieved by an extraction solution containing a mild antioxidant (Table 3) combined with Ellman’s reagent for the derivatization of GSH. The pH of this solution is in the acidic range and compares well with previously reported optimized conditions for the detection of NADH and NADPH [28]. Importantly, this method does not compromise detection of the majority of polar metabolites. Users may still wish to use the other buffers presented here depending on their interest in certain metabolites and according to the data presented here (Table 1 and Appendix A).

The comparison we present here between three different commercially available HILIC columns provides a comprehensive dataset that can help users decide on the optimal column for their needs (Appendix A). Although we achieved the broadest coverage using ZIC-pHILIC chromatography, we recognize that our comparison is not exhaustive and that further conditions or set of small molecules can be found that showcase the superiority of one column over another. However, the metabolites we have focused on here cover broadly major metabolic pathways in mammalian cells such as central carbon metabolism, TCA cycle and the ETC. Finally, the optimized ionization conditions reported here will be broadly applicable to most Thermo model MS instruments, like orbitrap and triple quad mass spectrometers, which are widely used in the metabolomics community. 

With the data we have presented here, we aim to facilitate successful study of the redox state of mammalian cells and tissues by applying metabolite profiling by LC-MS. We provide guidelines for choice of chromatography, MS method, extraction conditions, and extraction protocols and share the raw data of the detection of our full metabolite library to facilitate the planning of experiments that focus on metabolites that are not emphasized in this manuscript. The requirement for a robust method for detection of redox metabolites in parallel to other metabolites of interest is underlined by recent key publications that revealed the importance of profiling cellular redox state in various fields [4,5,6,7,8,9]. Our method can be applied to address a wide variety of biological questions that involve oxidative stress or redox imbalance and can resolve a difficulty to directly measure redox metabolites in the context of global metabolic state.

## 4. Materials and Methods

### 4.1. Animal Care, CSF and Organ Collection

All animal care and experimental procedures were approved by the Institutional Animal Care and Use Committees of Boston Children’s Hospital. The Committee for Animal Care at Boston Children’s Hospital approved all animal procedures carried out in this study under protocol number 19-07-3936. Mouse strain used was C57BL/6. Pure CSF samples were collected from the cisterna magna [39]. Blood samples were collected from the retromandibular vein. The samples were coagulated and centrifuged. Liver and kidney were collected and flash frozen. Tissue chunks were cut on a glass plate while kept chilled on top of dry ice.

### 4.2. Cell Culture and Treatments

K562 cells used in this manuscript were authenticated by short tandem repeat analysis and tested negative for mycoplasma. Cells were cultured in RPMI (Genesee Scientific) up to a density of 2 Million cells per mL. For redox chemical treatment experiments, cells were seeded at 1 Million cells per mL cell density in 6-well plates and drugs were added for 4 h at the following concentrations: methotrexate: 5 µM; oligomycin: 80 µg/mL; H_2_O_2_: 1 mM; diamide: 0.5 mM; DMSO, which served as control: 0.6 µL per 1 mL of cell culture media (equivalent to volume used for oligomycin). 

### 4.3. Sample Preparation for LC-MS Analysis of Polar Metabolites from Tissues or Cultured Cells (with Considerations for NADH and NADPH Detection)

Metabolites were quenched as quickly as possible while working with the samples at low temperatures. Cells were handled at 4 °C or on dry ice, extraction buffer was pre-chilled at −20 °C. Samples were analyzed by LC-MS on the same day of extraction (if impractical, best alternative is to store dried samples at −80 °C). Unless indicated otherwise, 1 million cells or about 2 mg of tissue was extracted per condition and a minimum of three replicates per condition was used in each experiment. K562 cells that are non-adherent, were collected by brief centrifugation at 4 °C using a table-top centrifuge (4500 rpm, 1.5 min) and washed briefly in 0.9% NaCl (high grade salt and LC-MS-grade water Fisher Scientific W6500 or Sigma Aldrich 1.15333). 300 µL of prechilled extraction buffer were added per sample. In the cases when Buffer C+Ellmans was used, 240 µL solution 1 (methanol component, containing isotopically labelled GSH standard) was added first, then immediately after 60 µL solution 2 containing Ellman’s reagent was added. This assured simultaneous derivatization of both endogenous as well as standard GSH. For tissues—chunks were crushed using a hand-held homogenizer (VWR 47747-370) with several pulses while keeping the samples on ice; 300 µL of prechilled extraction buffer was used per 2 mg of tissue.

For sample extraction, samples were vortexed briefly (10 sec) and sonicated for 3 min in a 4 °C water bath. Samples were then centrifuged for 10 min, 4 °C, at maximum speed on a benchtop centrifuge (Eppendorf) and the cleared supernatant was transferred to a new tube. Samples were dried using a nitrogen dryer while on ice, and special attention was given to minimize the time of drying and to not let samples idle in the dryer (Reacti-Vap™ Evaporator, Thermo Fisher Scientific, TS-18826) once the drying process was completed. Needles were continuously adjusted to the surface of the liquid as the samples evaporated to expedite the drying process. Samples were reconstituted in 30 µL LC-MS-grade water by brief sonication in a 4 °C water bath. Extracted metabolites were spun for 2 min at maximum speed on a bench-top centrifuge and cleared supernatant was transferred to LC-MS micro vials (National Scientific, C5000-45B). A small amount of each sample was pooled and serially diluted 3- and 10-fold to be used as quality controls throughout the run of each batch.

#### 4.3.1. Extraction buffer A

Acetonitrile:methanol:water in the ratio of 40:40:20, supplemented with 0.1 M formic acid and isotopically-labelled internal standards (17 amino acids and reduced glutathione, Cambridge Isotope Laboratories, MSK-A2-1.2 and CNLM-6245-10).

#### 4.3.2. Extraction buffer B

Eighty percent LC-MS-grade methanol, 20% 25 mM Ammonium Acetate and 2.5 mM Na-Ascorbate prepared in LC-MS water and supplemented with isotopically labelled internal standards (17 amino acids and isotopically labelled reduced glutathione, Cambridge Isotope Laboratories, MSK-A2-1.2 and CNLM-6245-10).

#### 4.3.3. Extraction buffer C and C + Ellman’s

Solution 1: 100% LC-MS Methanol

Solution 2: 25 mM Ammonium Acetate and 2.5 mM Na-Ascorbate in LC-MS water supplemented with isotopically labelled reduced glutathione and isotopically labelled internal standards (17 amino acids and reduced glutathione, Cambridge Isotope Laboratories, MSK-A2-1.2 and CNLM-6245-10).

Ellman’s reagent (5,5′-Dithiobis(2-nitrobenzoic acid),D8130, Sigma Aldrich): 20 mM in “Solution 2”. Final composition is 80% solution 1 and 20% solution 2. 

Samples were vortexed briefly (10 sec) and sonicated for 3 min in a 4 °C water bath. Samples were then centrifuged for 10 min, 4 °C, at maximum speed on a benchtop centrifuge (Eppendorf) and the cleared supernatant was transferred to a new tube. Samples were dried using a nitrogen dryer while on ice, and special attention was given to minimize the time of drying and to not let samples idle in the dryer (Reacti-Vap™ Evaporator, Thermo Fisher Scientific, TS-18826) once the drying process was completed. Needles were continuously adjusted to the surface of the liquid as the samples evaporated to expedite the drying process. Samples were reconstituted in 30 µL LC-MS-grade water by brief sonication in a 4 °C water bath. Extracted metabolites were spun for 2 min at maximum speed on a bench-top centrifuge and cleared supernatant was transferred to LC-MS micro vials (National Scientific, C5000-45B). A small amount of each sample was pooled and serially diluted 3- and 10-fold to be used as quality controls throughout the run of each batch.

### 4.4. Sample Preparation for LC-MS Analysis of Polar Metabolites from CSF (with Special Considerations for NADH and NADPH Detection)

CSF was collected by puncture of the cisterna magna with a glass capillary [39] and flash-frozen for further analysis. Per condition, 3 µL of precleared CSF were extracted by brief sonication in 200 µL of the indicated extraction buffers. After centrifugation for 10 min at maximum speed on a benchtop centrifuge (Eppendorf) the cleared supernatant was dried using a nitrogen dryer and reconstituted in 30 µL water by brief sonication. Extracted metabolites were spun again and cleared supernatant was transferred to LC-MS micro vials. A small amount of each sample was pooled and serially diluted 3- and 10-fold to be used as quality controls throughout the run of each batch.

### 4.5. Chromatographic Conditions for LC-MS

#### 4.5.1. ZIC-pHILIC Chromatography

One milliliter of reconstituted sample was injected into a ZIC-pHILIC 150 × 2.1 mm (5 µm particle size) column (EMD Millipore) operated on a Vanquish™ Flex UHPLC Systems (Thermo Fisher Scientific, San Jose, CA, USA). Chromatographic separation was achieved using the following conditions: buffer A was acetonitrile; buffer B was 20 mM ammonium carbonate, 0.1% ammonium hydroxide in water; resulting pH is around 9 without pH adjustment. Gradient conditions we used were: linear gradient from 20% to 80% B; 20–20.5 min: from 80% to 20% B; 20.5–28 min: hold at 20% B at 150 mL/min flow rate. The column oven and autosampler tray were held at 25 °C and 4 °C, respectively. 

#### 4.5.2. Accucore-HILIC Chromatography

One milliliter of reconstituted sample was injected into a Thermo Fisher Scientific™ Accucore™ 150 Amide HILIC (150 × 3 mm, 2.6 µm particle size; Thermo Fisher Scientific) operated on a Vanquish™ Flex UHPLC Systems (Thermo Fisher Scientific, San Jose, CA, USA). Chromatographic separation was achieved using the following conditions: buffer A was acetonitrile; buffer B was 20 mM ammonium carbonate, 0.1% ammonium hydroxide in water; resulting pH is around 9 without pH adjustment. Gradient conditions were: linear gradient from 20% to 80% B; 20–20.5 min: from 80% to 20% B; 20.5–28 min: hold at 20% B at 320 mL/min flow rate. The column oven and autosampler tray were held at 35 °C and 4 °C, respectively. 

#### 4.5.3. LUNA-NH2 Chromatography

One milliliter of reconstituted sample was injected into a Luna^®^ 3 µm NH2 100 Å, LC Column (150 × 2 mm, 3 µm particle size; Phenomenex, 00F-4377-B0) operated on a Vanquish™ Flex UHPLC Systems (Thermo Fisher Scientific, San Jose, CA, USA). Chromatographic separation was achieved using the following conditions: buffer A was acetonitrile; buffer B was 5 mM ammonium acetate and 0.2% ammonium hydroxide in water; resulting pH is around 9 without pH adjustment. Gradient conditions were: 20 min linear gradient from 10% to 90% B; 20–25 min hold at 90% B; 25–26 min from 90% to 10% B; 26–34 min hold at 10% B at 250 mL/min flow rate. The column oven and autosampler tray were held at 30 °C and 4 °C, respectively

### 4.6. Orbitrap Conditions for Targeted Analysis of Polar Metabolites

MS data acquisition was performed using a QExactive benchtop orbitrap mass spectrometer equipped with an Ion Max source and a HESI II probe (Thermo Fisher Scientific, San Jose, CA, USA) and was performed in positive and negative ionization mode in a range of m/z = 70–1000, with the resolution set at 70,000, the AGC target at 1 × 10^6^, and the maximum injection time (Max IT) at 20 msec. For tSIM scans, the resolution was set at 70,000, the AGC target was 1 × 10^5^, and the max IT was 100 msec. For PRM scans, the resolution was set at 17,500, the AGC target was 1 × 10^5^, and the max IT was 20 msec. The following inclusion list and energies were used (Table 6):

### 4.7. Data Analysis and Statistics

Relative quantitation of polar metabolites was performed with TraceFinder 4.1 (Thermo Fisher Scientific, Waltham, MA, USA) using a 5 ppm mass tolerance and referencing an in-house library of chemical standards (Table 1 and Appendix A). Pooled samples and fractional dilutions were prepared as quality controls and injected at the beginning and end of each run. In addition, pooled samples were interspersed throughout the run to control for technical drift in signal quality as well as to serve to access the coefficient of variability (CV) for each metabolite. Data normalizations were performed in two steps; 1. Integrated peak area signal from internal standards added to extraction buffers were mean-centered (for every standard, peak area was divided by the mean peak area of the set) and averaged across samples; samples were divided by the resulting factor, thus normalizing for any technical variability due to MS-signal fluctuation or pipetting and sample injection errors (usually withing 10% variability). 2. Normalization for biological material was based on detected metabolites as follows: CV values (based on pooled sample re-injections) and coefficient of determination (RSQ) (based on linear dilutions of pooled sample) were calculated per metabolite, metabolites with <30% CV and >0.95 RSQ were mean-centered and averaged across samples. Samples were then divided by the resulting factor (biological normalizer), thus accounting for any global shift in metabolite amounts due to differences in biological material. When different cell numbers were used, normalization for biological material was performed as described above but the biological normalizer was divided by the fold change in cell number to preserve the global differences in total metabolite amounts.

For downstream MetaboAnalyst-based statistical or pathway analysis, the data were either Log transformed and Pareto scaled [40] or normalized to control conditions where indicated. All heatmap, PCA, or PLSDA analysis were performed using the MetaboAnalyst online platform. Individual one-way Anova and t-tests were performed in Prism software. Multiple comparisons correction and false discovery rate tests were based on the two-stage step-up method of Bejamini, Krieger, and Yekuteili (as recommended within Prism software).

### 4.8. Data Deposition Information 

All data generated as part of this method development is deposited at Metabolomics Workbench doi: 10.21228/M81D6M, project ID: PR001127.

## Figures and Tables

**Figure 1 metabolites-11-00313-f001:**
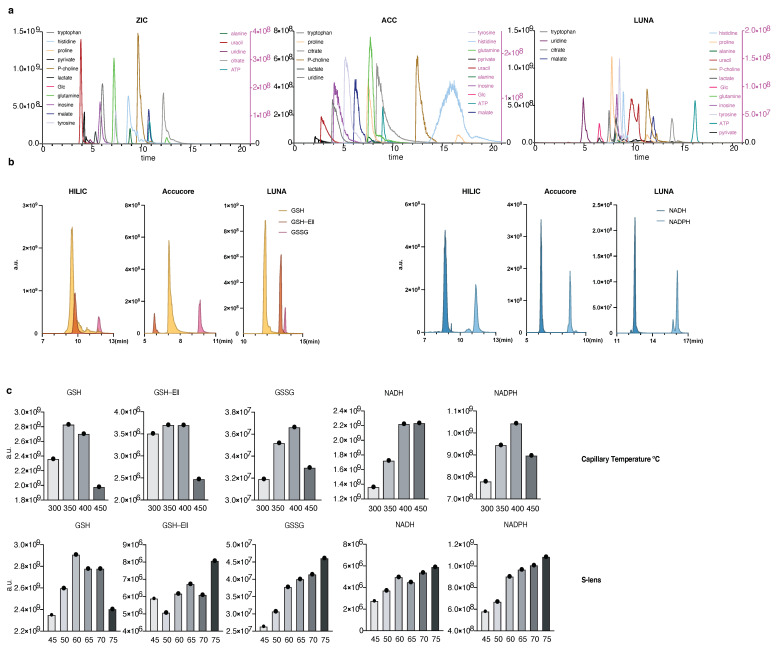
A method to quantify redox metabolites by LC-MS; (**a,b**) Retention times and sensitivity comparison between three HILIC-based chromatography methods. Chromatographic runs were carried out on either ZIC-pHILIC (ZIC), Accucore-NH2 (ACC), or LUNA-NH2 (LUNA) with 20 min, 20 min, or 25 min linear gradients, respectively. Overlaid peaks are shown for the indicated range of (**a**) polar metabolite standards (see Appendix A for further details) and (**b**) GSH, GSH derivatized with Ellman’s (GSH-Ell), GSSG, NADH, and NADPH. Redox metabolites were diluted in 25 mM Ammonium Acetate and 2.5 mM Na-Ascorbate in water. Standards were dissolved in extraction buffer B (as detailed in Section 2.2); (**c**) Optimization of HESI parameters on orbitrap mass spectrometer for the indicated metabolites using ZIC-pHILIC chromatography and standards. Graphs represent integrated areas of chromatographic peaks under changing parameters for capillary temperature or S-lens; (**d**) Limit of detection and linearity for individual redox molecules. Presented are the average values and standard deviation.

**Figure 2 metabolites-11-00313-f002:**
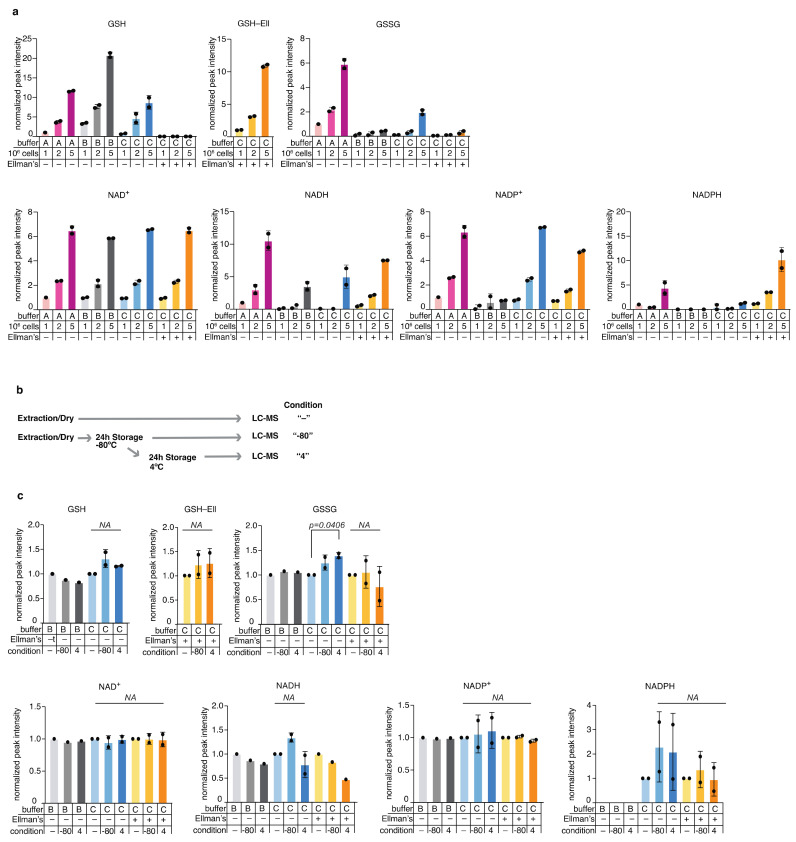
Optimization of redox metabolite detection in mammalian cells; (**a**) Comparison of metabolite detection in samples extracted from the mammalian cell line K562 using three extraction buffers (A, B, and C), with increasing number of cells (1, 2, or 5 million cells). Values were normalized to Buffer A/1 Million cells condition and present the average and standard deviation of two experimental replicates, each with technical duplicates (except for buffer A, 1M condition, which was measured once with a technical duplicate; (**b**) A scheme of the metabolite stability tests performed of various storage conditions: One million K562 cells were extracted immediately upon harvest and either analyzed by LC-MS immediately (labelled “-“), or analyzed after they were stored for 24 h at −80 °C (labelled “−80”), or analyzed following reinjection that followed additional 24 h incubation period at 4 °C (labelled “4”); (**c**) Stability of redox metabolites extracted from 1M K562 cells in different storage conditions. Each storage condition was analyzed following extraction in the extraction buffers B and C. Values were normalized for each buffer to the corresponding “-” condition that involved no long storage. Presented are the average values and standard deviation of two independent experiments each with technical triplicates.

**Figure 3 metabolites-11-00313-f003:**
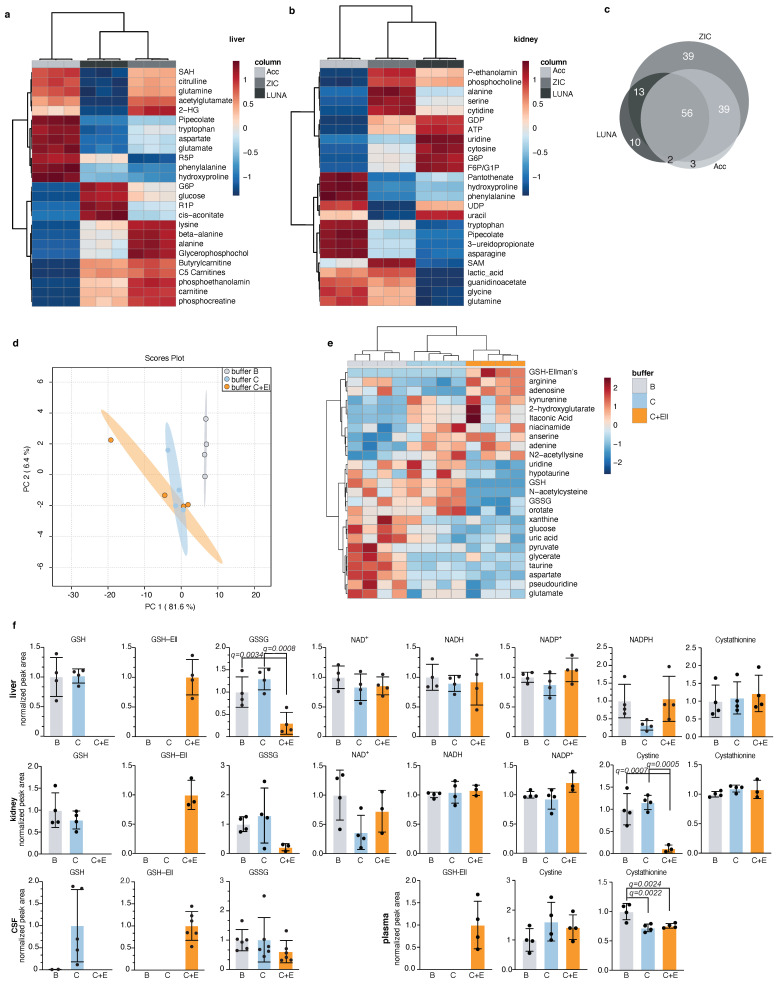
Application of the redox metabolite detection method for mammalian tissues; (**a**,**b**) Top 25 differentially detected metabolites between three chromatographic methods in mouse liver (**a**) and kidney (**b**) presented in a heatmap for each organ. Each replicate is a tissue chunk from an independent mouse, extracted in buffer C+Ell and then split in three and run on the different chromatographies. ZIC: ZIC-pHILIC, Acc: Accucore-NH2, LUNA: LUNA-NH2; (**c**) Venn diagram of total detected metabolites in liver and kidney between the three chromatographic methods from (**a**,**b**) and the overlap between them. Only metabolites passing 30% CV cutoff were compared; (**d**) Global PCA analysis of polar metabolites detection in mouse liver samples extracted by three different buffers (B, C, C plus Ellman’s) and detected by LC-MS using the ZIC-pHILIC column; Each replicate represents a liver chunk from an independent mouse, subdivided evenly between three tubes and extracted using the different buffers. This experiment was performed twice and a representative replicate is shown; (**e**) Top 25 differentially detected metabolites in mouse liver samples from (**d**); (**f**) Detection of redox metabolites from mouse liver, kidney, CSF, and plasma extracted in three different buffers (B, C, and C plus Ellman’s). These experiments were performed twice (except for plasma) and a representative replicate is shown. Each dot represents a measurement from an independent mouse, mean and standard deviation are indicated; statistical significance was determined using Anova with correction for multiple comparisons by false discovery rate correction. Only significant q-values are indicated (except for GSH and GSH-Ell levels, which were excluded from analysis).

**Figure 4 metabolites-11-00313-f004:**
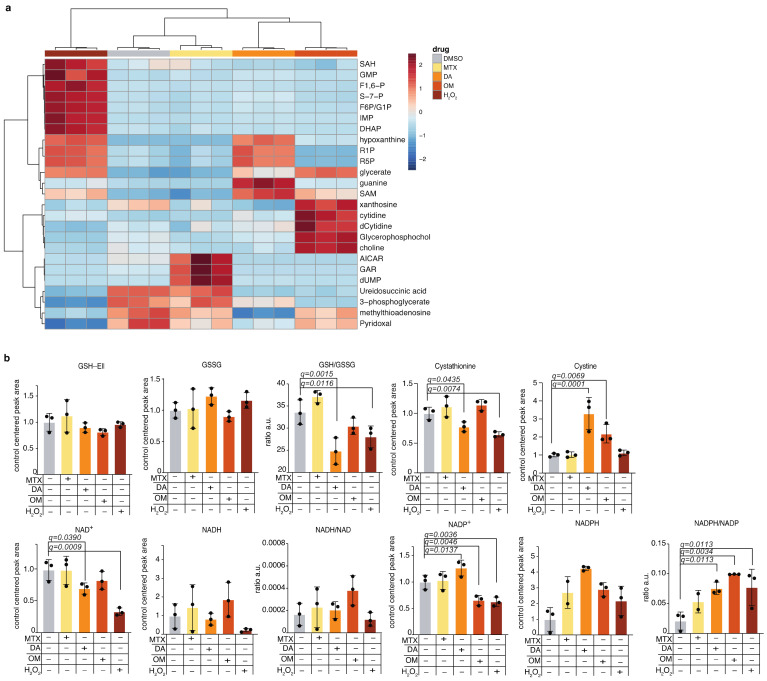
Application of the redox metabolite detection method for profiling redox state following pharmacologic perturbations of redox balance in cells; (**a**) K562 cells were treated with drugs that perturb the cellular redox balance and with H_2_O_2_, using the following doses for 4 h: methotrexate: 5 µM; oligomycin: 80 µg/mL; H_2_O_2_: 1 mM; diamide: 0.5 mM; DMSO, which served as control: 0.6 µL per 1mL of cell culture media. The drugs used are diamide (DA), oligomycin (OM), and methotrexate (MTX). Global heatmap analysis of the top 25 differentially-changed polar metabolites is presented; (**b**) Detected levels of redox metabolites in the treated cells. Mean and standard deviation of three technical replicates are presented; statistical significance was determined using Anova with correction for multiple comparisons and false discovery rate. Only significant q-values are indicated.

**Table 1 metabolites-11-00313-t001:** Retention times and HESI polarity mode for metabolite standards shown in Figure 1. Three chromatographic column separations are indicated. A full list of characterized metabolites can be found in Appendix A.

Standard Name	Exact Mass	Neg m/z	Pos m/z	Mode	RT (min.) ZIC-pHILIC	RT (min.) Accucore HILIC	RT (min.) LUNA-NH2
Alanine	89.0477	88.0404	90.055	pos	8.85	10.4	8.1
ATP	506.9958	505.9885	508.003	neg	10.83	9.96	15.75
Citric acid	192.027	191.0197	193.0343	neg	12.23	9.7	13.42
Glucose	180.0634	179.0561	181.0707	neg	9.33	7.5	6.46
Glutamine	146.0691	145.0619	147.0764	pos	9.13	9.12	8.92
Histidine	155.0695	154.0622	156.0768	pos	9.31	12	9.47
Inosine	268.0808	267.0735	269.088	pos	5.98	4.3	8.69
Lactic acid	90.0317	89.0244	91.039	neg	5.41	2.82	6.46
Malic acid	134.0215	133.0142	135.0288	neg	10.74	6.6	11.56
Choline phosphate	183.066	182.0588	184.0733	pos	10.17	14.66	10.95
Proline	115.0633	114.0561	116.0706	pos	7.22	9.04	7.67
Pyruvic acid	88.016	87.0084	89.0233	neg	4.32	2.34	N/A
Tryptophan	204.0899	203.0826	205.0972	pos	6.25	4.08	7.61
Tyrosine	181.0739	180.0666	182.0812	pos	6.16	5.85	8.71
Uracil	112.0273	111.02	113.0346	neg	4.1	3.67	3.66
Uridine	244.0695	243.0623	245.0768	neg	4.81	5.38	5.13
Glutathione	307.0838	306.0765	308.0911	pos	9.64	12.80	11.13
Oxidized glutathione	612.1520	611.1447	613.1592	pos	11.95	15.02	13.55
GSH-Ellman’s	504.0625	503.0552	505.0698	neg	9.85	12.07	12.02
NAD^+^	664.1170	662.1019	664.1164	pos	9.36	12.69	10.99
NADH	665.1248	664.1175	666.1320	pos	8.85	12.22	12.60
NADP+	744.0833	742.0682	744.0827	pos	11.13	14.95	14
NADPH	745.0911	744.0838	746.0984	pos	11.44	14.15	16.24

**Table 2 metabolites-11-00313-t002:** Linear regression and goodness of fit (R-squared: R^2^; and standard deviation of the residuals: Sy.x; Number of replicated measurements: N) for redox metabolites on three different columns.

Standard Name	ZIC			ACC			LUNA		
	R^2^	Sy.x	N	R^2^	Sy.x	N	R^2^	Sy.x	N
GSH	0.8796	0.4638	4	0.9160	0.3891	2	0.9979	0.0656	1
GSH-Ell	0.9682	0.1915	4	0.9919	0.1066	2	N/A	N/A	-
GSSG	0.9860	0.1571	3	0.9200	0.3288	2	0.9985	0.0354	1
NADH	0.9506	0.3078	2	0.9277	0.3473	2	0.9804	0.1423	2
NADPH	0.9688	0.2285	2	0.9009	0.3987	2	0.9578	0.2225	2

**Table 3 metabolites-11-00313-t003:** Extraction buffer composition.

Buffer	Composition	Reference
A	40:40:20 acetonitrile:methanol:water, 0.1 M formic acid	[28] Lu W, et al. Antioxid Redox Signal. 2018 Jan 20;28(3):167–179.
B	80% methanol, 20% 25 mM Ammonium Acetate, 2.5 mM Na-Ascorbate	[29] Chen L, et al. Anal Bioanal Chem. 2017 Oct;409(25):5955–5964.
C	Solution1: 100% Methanol Solution2: 25 mM Ammonium Acetate, 2.5 mM Na-Ascorbate Use 80% Solution 1/20% Solution 2	
C + Ellman’s	20 mM Ellman’s in Solution 2	[27] Ellman, G.L., *Tissue sulfhydryl groups.* Arch Biochem Biophys, 1959. **82**(1): p. 70–7

**Table 4 metabolites-11-00313-t004:** Linearity and reproducibility for four buffer comparisons. R squared (R^2^) values were calculated based on linearity of fit between relative change and cell count. Coefficient of variation (CV) was calculated considering the variability between two biological replicates for three cell counts. Values in cells represent [R^2^; CV(%)].

	Buffer	A	B	C	C+Ell
Metabolite					
GSH		0.9999; 5.36	0.9999; 5.96	0.9254; 25.98	0.9983; 3.44
GSSG		0.9999; 8.76	0.9988; 56.18	0.9828; 17.93	0.9973; 23.60
NAD+		0.9999;3.52	0.9999; 7.43	0.9996; 3.75	0.9999; 5.57
NADP+		0.9970; 5.72	0.7124; 78.45	0.9986; 5.93	0.9984; 3.85
NADPH		0.8614; 28.18	N/A; N/A	0.7407; 63.25	0.9999; 11.16

**Table 5 metabolites-11-00313-t005:** Linearity and reproducibility for three HILIC columns. R squared (R^2^) values per metabolite were calculated based on linearity of fit between measured relative change in pooled samples and corresponding dilution factor. Coefficient of variation (CV) per metabolite was calculated from re-injections of pooled samples. Indicated are number of metabolites which pass specified thresholds.

Column	Metabolites—R^2^	Metabolites—CV (%)
ZIC	121—0.97 122—0.95	152—30% 141—20%
Acc	69—0.97 69—0.95	78—30% 58—20%
LUNA	26—0.97 39—0.95	79—30% 45—20%

**Table 6 metabolites-11-00313-t006:** Inclusion list for PRM analysis of specified metabolites.

Name	m/z	Polarity	NCE	RT Range (min)
GSH-Ellman’s	503.0552	negative	20, 40, 60, 80	8–12
GSH-13C2-15N-Ellman’s	506.0580	negative	20, 40, 60, 80	8–12
GSH	308.0911	positive	20, 40, 60, 80	8–12
GSH-13C2-15N	309.0802	positive	20, 40, 60, 80	8–12
GSSG	613.1592	positive	20, 40, 60, 80	10–14
NADPH	746.0984	positive	20, 40, 60, 80	9–14
NADP+	744.0827	positive	20, 40, 60, 80	8–13
NADH	666.1320	positive	20, 40, 60, 80	6–11
NAD+	664.1164	positive	20, 40, 60, 80	6–11

## Data Availability

The data presented in this study are openly available in Metabolomics Workbench at doi: 10.21228/M81D6M, project ID: PR001127.

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
