# Peer review of "Redox Metabolism Measurement in Mammalian Cells and Tissues by LC-MS"

_metabolites, 2021, doi:10.3390/metabo11050313_

Round 1

Reviewer 1 Report

The submitted manuscript describes development of a method for measurement of some key metabolites representing the redox state of mammalian cells. Sample preparation procedures and detection parameters are described, and some applications are presented.

I have following comments and questions to the submitted material.

1) Certainly, the metabolites under investigation are polar ones, therefore, HILIC is likely to be most convenient type of separation for them. Nevertheless, it is interesting to know whether any revesed-phase columns can be used for the same purpose. Nowadays, more and more studies appear using different columns with either C18 sorbent (endcapped or not) or aromatic-containing stationary phase. Have the authors ever tried to utilize a reversed-phased chromatography to detect the metabolites?

2) The fig 1 seems to be overloaded. I suggest removing the sub-figure 1d and representing the data as a table.

3) Page 5, section 2.2: I think that the section title does not reflect its material. May be, better use like "Optimization of mammalian cells sample preparation method for redox metabolite detection"

4) I suggest rearrangement of the Table 3 as follows:

Metabolite

/buffer

A B C C+Ellman's
GSH 0.9999; 5.36 0.9999; 5.96    
GSSG 0.9999; 8.76 0.9988; 56.18    
NAD+        
NADP        
NADPH        

Values in cells represent [R2; CV(%)].

5) Section 3 (Discussion): The use of developed protocol for the metabolites detection in CSF is discussed. Nevertheless, I suppose that plasma or serum as a biological matrices are of much more interest as they are used in most of metabolomics studies. Were there any preliminary studies carried out to measure the metabolites in these matrices and, thus, to show its applicability?

6) Section 4 (Materials and methods): Please number the subsections.

7) Section 4, Chromatographic condition: 

It is necessary to mention that water was the solvent in the eluent B. Also, what was the pH value of the prepared eluent? Was it adjusted to a certain value or the eluent was just prepared from standard solutions? Flow rate also should be given for each column, as they have different internal diameter.

8) References should be formatted in accordance with the Journal's rules.

Reviewer 2 Report

The article is devoted to the optimization of the method for determining markers of oxidative stress in mammalian cells and tissues. The article is an original research, clearly and competently planned, logically stated. Question to the authors: why did they limit themselves to oxidized and reduced glutatitone and NADPH, NADH? Of thiols, homocysteine and cysteine are also interesting. Table 1 shows a list of the compounds to be determined, which includes tyrosine and tryptophan. Why is there no methionine? In general, I think the article is suitable for publication. 

Reviewer 3 Report

In the current manuscript by Petrova and Colleague have compared analytical accuracy of three commercially available LC-MS based platform viz. Accucore Amide HILIC (Thermo-Fisher Scientific), LUNA-NH2 (Phenomenex), and SeQuant ZIC-pHILIC (Millipore-Sigma), by performing quantitative analysis of reduced and oxidized glutathione, NAD+, NADH, NADP+, NADPH, amino acids, etc. in cell line and biospecimen. Further, the authors have also compared three previously reported metabolites extraction methods suitable for the detection of redox-sensitive metabolites on above mentioned three LC-MS platforms. 
The current research topic is quite important in metabolomics studies especially oxidation-reduction in the context of metabolism. Intracellular redox changes have profound effects on cellular metabolism. I have few questions as follows:  
Figure 1b why NADPH RT peaks in histogram changed which substantially moved from ZIcC to LUNA?  Need to explain in results.  Did the authors determine the pH of buffer A before preparation/extraction? Knowing that slight change in pH of extraction buffer affects the stability of NADPH. 
Figure 2a; with the addition of Ellman (a sulfhydryl reagent) in buffer C, there is almost no GSH and no change in GSSG either? though I do see NADPH increased? since its recycling assay GSSG -GSH, I am wondering if the authors compared data with another enzymatic assay of GSH and validated the data?
Figure 2C and Figure 2A: compare NADPH in buffer C. is it quite interesting that lysate preserved at 40C and -800C showed increased NADPH and decreased NADH compared to fresh? Since NADPH is readily oxidized. The authors need to discuss these points. 

Figure 2A and Figure 2C: GSSG: addition of Ellman reagent reduced the GSSG level (figure 2A), on the contrary GSSG detected in fresh/frozen lysate as shown in figure 2C. why? is there any difference between these two figures except storage? 
Figure 3: authors can provide a Venn diagram of global metabolites detected on three LC-MS platforms and annotate the number of overlapped and unique metabolites in each platform. 
Method: Line 326; authors can briefly describe that why they used Ellman's reagent with buffer C in the method section. 
I found the data is quite impressive. However, I do feel that authors need to explicitly elaborate their findings in results.

The discussion part is very suboptimal and poorly discussed their findings with suitable reference. I also recommend authors to highlight their key findings from each figure, which would be critically important in the selection of suitable methods and determination of redox-mediated metabolic change and metabolomics assays.   
Minor: 
Bar diagram in Figures: Authors need to change the color of bars from white to other colors and borders, which should be clear, visible, and easy to read. 
Figure 1b: I would suggest authors to replot the AUC histogram of NAD+/NADH/NADP/NADPH overlay in a separate figure from GSH and GSSG which also need to clearly visualize into the histogram with dark borders / light color/transparent background. Authors can reduce the X-axis range from 5 - 20.   
